# Catestatin—A Potential New Therapeutic Target for Women with Preeclampsia? An Analysis of Maternal Serum Catestatin Levels in Preeclamptic Pregnancies

**DOI:** 10.3390/jcm12185931

**Published:** 2023-09-12

**Authors:** Pilar Palmrich, Nawa Schirwani-Hartl, Christina Haberl, Peter Haslinger, Florian Heinzl, Harald Zeisler, Julia Binder

**Affiliations:** Department of Obstetrics and Gynecology, Division of Obstetrics and Feto-Maternal Medicine, Medical University of Vienna, 1090 Vienna, Austria; pilar.palmrich@meduniwien.ac.at (P.P.); nawa.schirwani-hartl@meduniwien.ac.at (N.S.-H.); christina.haberl@meduniwien.ac.at (C.H.); peter.haslinger@meduniwien.ac.at (P.H.); florian.heinzl@meduniwien.ac.at (F.H.); harald.zeisler@meduniwien.ac.at (H.Z.)

**Keywords:** catestatin, hypertensive disorders of pregnancy, preeclampsia, biomarkers, maternal and neonatal morbidity and mortality, chromogranin A

## Abstract

Background: Catestatin has been identified as an important factor in blood pressure control in non-pregnant adults. A possible impact on the development of hypertensive disorders of pregnancy has been indicated. Data on catestatin levels in pregnancy are scarce. The aim of this study was to investigate a potential association of maternal serum catestatin levels to the pathogenesis of preeclampsia. Methods: We evaluated serum catestatin levels of 50 preeclamptic singleton pregnancies and 50 healthy gestational-age-matched pregnancies included in the obstetric biobank registry of the Medical University of Vienna. Receiver operating characteristic curves and logistic regression models were performed to investigate an association between catestatin levels and development of preeclampsia. Results: Catestatin levels were significantly decreased in women with preeclampsia compared to healthy controls (median CST: 3.03 ng/mL, IQR [1.24–7.21 ng/mL] vs. 4.82 ng/mL, IQR [1.82–10.02 ng/mL]; *p* = 0.010), indicating an association between decreased catestatin values and the development of preeclampsia. There was no significant difference in catestatin values between early-onset preeclampsia and late-onset preeclampsia. Modelling the occurrence of preeclampsia via logistic regression was improved when adding catestatin as a predictive factor. Conclusions: Decreased serum catestatin levels are associated with the presence of preeclampsia. Further investigations into the diagnostic value and possible therapeutic role of catestatin in preeclampsia are warranted.

## 1. Introduction

Preeclampsia is one of the leading causes of maternal and neonatal morbidity and mortality, complicating 2–8% of all pregnancies worldwide [1,2]. It is traditionally defined as hypertension in pregnancy combined with proteinuria or other maternal end-organ dysfunction, such as renal insufficiency, liver involvement or uteroplacental dysfunction [3]. Reducing the number and degree of serious maternal and fetal complications remains a challenge [4]. The underlying pathophysiology and pathomechanism leading to preeclampsia are not yet fully understood and have been the focus of research in recent decades [5,6].

Recently, the hypothesis of preeclampsia being a placental disorder mediated by inadequate trophoblast invasion was challenged. A number of studies have found evidence for short- and long-term cardiovascular changes in pregnancies affected by preeclampsia, suggesting a cardiovascular origin of the disease [7,8,9,10]. However, strategies of treating preeclampsia have not changed fundamentally in recent years. Increased blood pressure remains the main target of therapy trying to reduce maternal and neonatal morbidity and mortality by using antihypertensive agents. Current recommendations for antihypertensive agents include methyldopa, labetalol and nifedipine as first and second line agents [4,11]. However, evidence for better maternal and neonatal outcome is lacking [12].

In 1997, Mahata et al. [13] identified catestatin, a chromogranin A (CgA)-derived fragment peptide, which is released from chromaffin cells of the adrenal medulla and adrenergic neurons, as a potent inhibitor of catecholamine secretion [14]. Catestatin can act as a nicotinic cholinergic antagonist in chromaffin cells and inhibits its release by negative feedback mechanisms [15,16].

Catestatin has also been identified as a strong vasodilator via different pathways, including mast cell release of histamine [17]. A relevant vasoreactive effect of catestatin has also been shown in animal models. It has been demonstrated that CgA knockout mice developed high blood pressure, which could effectively be treated with replacement therapy of catestatin, indicating a potential novel target for treatment of hypertension [18]. Catestatin has not only been identified as an important factor in blood pressure control, it also shows direct cardiovascular effects as an inotropic and lusitropic agent [14,15,16]. Human studies have also shown that circulating levels of catestatin are decreased in patients with chronic hypertension, especially early on in the development of the disease, suggesting a potential influence of reduced catestatin levels on the risk for the development of hypertension [16,17,18,19].

Additionally, catestatin has been found to share several features with angiogenic factors such as neuropeptide substance P and vascular endothelial growth factor, showing vasodilative activity [14]. Angiogenic biomarkers have been found to play an important role in the development of placental dysfunction and related diseases [20,21,22]. Various studies have demonstrated that an angiogenic imbalance in preeclampsia, reflected by excessive placental secretion of the circulating anti-angiogenic molecule, soluble fms-like tyrosine kinase-1 (sFlt-1) and decreased angiogenic placental growth factor (PlGF) levels in the maternal circulation, is present in preeclampsia [23,24]. A recent study by Bralewska et al. investigating the difference in placental concentrations of catestatin found significantly lower levels of catestatin in placentas of preeclamptic patients compared to normotensive controls, further indicating a possible impact on the development of hypertensive disorders of pregnancy (HDP) [25].

The aim of this study was to investigate serum levels of catestatin in women affected by preeclampsia and healthy controls to further elucidate a potential association with the pathogenesis of preeclampsia as a prerequisite for a possible therapeutic effect of catestatin in preeclampsia.

## 2. Materials and Methods

This study included a retrospective analysis of prospectively collected data as part of the obstetric biobank registry of the Department of Obstetrics and Feto-Maternal Medicine at the Medical University of Vienna (ethics approval number 1878/2018). Written informed consent was obtained for each patient for the collection of demographic and outcome data as well as for collection and storage of serum blood samples. The demographic, clinical and laboratory information were obtained as well from the obstetrical database Viewpoint (Viewpoint 5.6.8.428, Wessling, Germany), pseudonymized and stored in an electronic database, Scicomed. This study evaluated serum levels of catestatin in 50 patients diagnosed with preeclampsia and 50 healthy control subjects included in the obstetric biobank registry. Data collection began 1 November 2018 and lasted until November 2019. Inclusion criteria were as follows: singleton pregnancies with diagnosed preeclampsia. Preeclampsia was defined according to the revised criteria of the International Society for Hypertension in Pregnancy in 2014 (ISSHP) (3). Hypertension was defined as new-onset systolic blood pressure of ≥140 mm Hg and/or diastolic blood pressure of ≥90 mm Hg on 2 separate occasions ≥24 h apart. Significant proteinuria for a diagnosis of preeclampsia was confirmed with either protein/creatinine ratio ≥30 mg/mmol or ≥300 mg protein/24 h. Early-onset preeclampsia was defined as preeclampsia developing before 34 weeks of gestation [3]. Women with an age under 18 years, chronic hypertension, chronic renal disease, diabetes mellitus type I and II, autoimmune diseases, e.g., lupus erythematodes, inflammatory diseases or other conditions that might influence catestatin levels such as acute infections or recent surgeries or injuries as well as pregnancies with aneuploidy, genetic syndromes or major structural fetal anomalies were excluded. Small for gestational age (SGA) was defined as a birthweight below the 10th centile [26].

Patients with preeclampsia were recruited at time of diagnosis, and blood samples were obtained once at the time of study inclusion. The control group consisted of healthy pregnant women matched for gestational age who were recruited at their first antenatal clinic visit at the Department of Obstetrics and Feto-Maternal Medicine between 11+6 and 13 + 6 weeks of gestation and were followed up in four weekly intervals according to the biobank registry protocol. Blood samples were obtained at each visit and matched with blood samples of preeclamptic women according to gestational age at sampling. Blood was drawn by venipuncture, and the sample was stored in a collection tube without anticoagulants. Serum was processed according to the biobank protocol as well as stored in a −80 degree C freezer. Laboratory analysis was done by a biomedical technician using the human Catestatin Elisa Kit from Phoenix Pharmaceuticals Inc. (Burlingame, CA, USA). The analysis was done according to the manufacturer’s protocol, and additional measurements were obtained using the FLUOstar OPTIMA Microplate-reader (BMG Labtech, Ortenberg, Germany) at a wavelength of 450 nm. Each sample was measured twice to increase accuracy.

### Statistical Analysis

Numerical variables are represented via median and interquartile range (IQR), and comparisons between two groups for these variables were performed by using Wilcoxon rank sum tests. Categorical variables were summarized by absolute and relative frequencies, and Chi^2^ tests were used to compare groups.

To analyze a possible linear dependence of catestatin levels on gestational age, the Pearson correlation coefficient was calculated for the entire sample as well as the two subgroups (case and control). Furthermore, a receiver operating characteristic (ROC) curve was created to identify possible thresholds of catestatin to rule in or respectively rule out preeclampsia. Lastly, two different logistic regression models for the diagnosis of preeclampsia were calculated and then compared. *p* values ≤ 0.05 were considered statistically significant. Due to the exploratory character of the study, no multiple testing adjustment was applied.

Power analysis revealed the following: in case of the Chi^2^ test, under the assumptions of a sample size of 100, a significance level equal to 0.05 and a power of 0.8, we can detect effects of sizes 0.28, 0.31 and 0.33 for degrees of freedom 1, 2 and 3, respectively, which correspond to a medium effect size (0.3). In order to employ power analysis for the Wilcoxon test, we need to resort to Pitman Asymptotic Relative Efficiency (ARE) [27], since it is a non-parametric test. In relation to student’s *t* test, the ARE of the Wilcoxon test is at worst 0.864; hence, we analyzed the situation for the two-sided unpaired *t*-test under the following assumptions: sample size (per group) equal to 43, power equal to 0.8 and a significance level of 0.05. This yields an effect size of approximately 0.6, which again corresponds to a medium effect size (0.5). For our ROC curve, we calculated, again using the standard assumptions as well as our sample size, a power of 0.76.

Concerning logistic regressions, we tended to follow the (conservative) rule-of-10 as there is no power analysis for these models, which means one predictor per ten events (=preeclampsia diagnosis in our case). Therefore, we are confident in using up to five predictors per regression. Statistical analysis was performed using the statistical software R, version 4.2.1 (including packages pROC, ggplot2 and pwr) and SPSS 26.0 (SPSS Inc., Chicago, IL, USA).

## 3. Results

### 3.1. Patient Characteristics and Outcomes of the Study Population

The study included 50 women with preeclampsia and 50 healthy pregnancies matched for gestational age. The patient cohort comprised 34 women with early-onset preeclampsia and 16 women with late-onset preeclampsia. We compared baseline characteristics and pregnancy outcome variables between the case and the control group (Table 1). Maternal age (median age in controls: 29.50 years, IQR 26.00–34 vs. median age of 33.00 years, IQR 27.25–37.00; *p* = 0.0576) was lower in the control group compared to preeclamptic patients, though the difference was not statistically significant. BMI was significantly lower in the control group compared to women with preeclampsia (control group: 26.05, IQR 23.53–29.08; preeclampsia group: 29.40, IQR 25.88–34.05; *p* = 0.002). The preeclamptic group had a significantly larger number of nulliparous women (*n* = 32, 64%) compared to controls (*n* = 19, 38%); *p* = 0.0164. Mean arterial pressure at time of sampling was elevated in preeclamptic women (median MAP: 128 mmHg, IQR 114.25–135.5 mmHg) compared to controls (median MAP: 83.33 mmHg, IQR 76.67–87.33 mmHg; *p* < 0.0001). Gestational age at delivery was significantly lower in women diagnosed with preeclampsia (median: 32.79 weeks, IQR 28.93–36.1; vs. median: 39.71, IQR 38.57–40; *p* < 0.0001 for the control group). Birth weight was significantly lower in neonates born to preeclamptic women (*p* = 0.0191). Within the preeclampsia group, 31 of 50 women (62%) delivered a small for gestational age (SGA) neonate, while the percentage of SGA neonates within the control group was significantly lower at 4% (*p* < 0.0001). A total of 32 (96%) newborns within the case group were admitted to the neonatal intensive care unit (NICU), compared to no NICU admissions within the control group. Overall, one intrauterine death (IUD) and three neonatal deaths (NND) within one week after delivery occurred in the preeclampsia group, all of which were cases with early onset preeclampsia, while no IUD or neonatal death occurred within the control cohort (Table 1).

### 3.2. Catestatin Levels in Women with Preeclampsia

Serum catestatin levels were significantly lower in women with preeclampsia compared to healthy controls (median catestatin: 3.03 ng/mL, IQR [1.24–7.21 ng/mL] versus 4.82 ng/mL, IQR [1.82–10.02 ng/mL]; *p* = 0.010), as seen in Figure 1. We found no significant difference in catestatin values between early-onset preeclampsia and late-onset preeclampsia (early-onset preeclampsia, median catestatin: 1.63 ng/mL, IQR [1.04–6.58 ng/mL]; and for late-onset preeclampsia, 6.39 ng/mL, IQR [1.32–8.5 ng/mL]; *p* = 0.203). We calculated thresholds for ruling in preeclampsia (catestatin level of 0.975 ng/mL corresponding to a specificity of 0.96), respectively ruling out preeclampsia (catestatin level of 12.268 ng/mL corresponding to a sensitivity of 0.96), which are based on our computation of an ROC curve. Figure 2 shows said ROC curve, the blue area around the curve representing the 95% CI for the curve.

We performed a logistic regression analysis for the prediction of preeclampsia with the predictors catestatin levels, BMI, smoking and nulliparity. Best predictive accuracy could be achieved using a combined model with all four predictors (*p* < 0.001). While the predictors BMI, smoking and nulliparity show significance as solitary predictors as well (*p* < 0.05), catestatin as a predictor alone was not significant, though a tendency can be observed (*p* = 0.07). However, the model including catestatin showed superiority to a model without catestatin (predictors BMI, smoking and nulliparity) when comparing the respective Akaike information criterion (118 versus 119.4). This assertion also holds when examining the respective residual deviances (108 for the full model versus 111.4 without catestatin levels). Comparison of the residual deviances for the single predictors showed comparable values for the predictors catestatin, nulliparity and smoking (residual deviances of 3.4 versus 5.4 and 6.2); only BMI showed a higher impact as a predictor (residual deviance of 12.7).

## 4. Discussion

### 4.1. Summary of the Key Findings

This case control study evaluated maternal serum catestatin levels in women with preeclampsia compared to a healthy control cohort. To date, only few studies have assessed catestatin values in preeclampsia, while this was the first study to investigate their predictive capability. Furthermore, this is the first study to assess and to explore potential diagnostic cut-off values for catestatin in women with preeclampsia. Serum catestatin levels were significantly decreased in women with preeclampsia compared to healthy control patients, indicating an association between decreased catestatin values and the development of preeclampsia. There was no significant difference in catestatin values between early-onset preeclampsia and late-onset preeclampsia. In order to explore thresholds of catestatin levels to rule in and vice versa rule out preeclampsia, we created ROC curves in which we were able to calculate thresholds of 0.975 ng/mL for ruling in and 12.268 ng/mL for ruling out preeclampsia. However, due to the small sample size, these thresholds are strictly exploratory and require further analysis with larger sample sizes to be potentially identified as diagnostic cut-off values. Additionally, we compared logistic regression models with possible confounders for preeclampsia including BMI, smoking and nulliparity with and without catestatin levels as an additional predictor. The model including catestatin levels as an added predictive factor showed superiority compared to the model without catestatin, further indicating an association between decreased catestatin values and the development of preeclampsia.

### 4.2. Interpretation of the Study Findings and Comparison with Existing Literature

Various recent studies have compiled growing evidence for short- and long-term cardiovascular changes in pregnant women with HDP [5,7,8,9,10,28] implicating an underlying cardiovascular etiology rather than a placental origin of the disease and generating substantial grounds for new therapeutic approaches to the disease. As mentioned, it has been hypothesized that decreased circulating serum catestatin levels could potentially have an influence on the risk for development of hypertension [19,29]. To date, only a few studies evaluating maternal serum catestatin levels in pregnant women with preeclampsia have been reported [30,31]. A study by Tüten et al. found elevated serum catestatin levels in women with mild and severe preeclampsia compared to normotensive controls, contradictory to existing literature on catestatin levels in hypertensive patients [19,29,30]. Contrary to these results, we were able to show that serum catestatin levels were decreased in women with preeclampsia compared to gestational-age-matched control subjects, as expected. Catestatin decreases blood pressure via suppression of catecholamine release and stimulation of histamine release [29]. It acts as a vasodilator and has cardioprotective qualities as an inotropic and lusitropic agent [14,17,19]. An increase in catestatin levels, therefore, does not seem to be in agreement with the pathomechanisms described above. Our results are in line with data showing that circulating levels of catestatin are decreased in patients with chronic hypertension [14,16,17,19], catestatin being a key player in regulation of blood pressure [29]. Furthermore, our findings are consistent with a recent study by Özalp et al., which also showed decreased serum catestatin levels in preeclamptic pregnancies [31]. We were not able to demonstrate differences in catestatin levels in early- and late-onset preeclampsia, reflecting that the disease is associated with cardiac impairment irrespective of the gestational age of onset. This is also in line with literature published recently describing cardiac changes in both early-onset and late-onset preeclampsia [9,10,28,32]. Consistent with human findings, the role of catestatin as a blood pressure regulator has been demonstrated in animal models. A lack of catestatin in CgA knockout mice caused elevated blood pressure values, which could effectively be treated by supplementation with catestatin, thus indicating potential therapeutic effects of catestatin for treatment of hypertension [18].

### 4.3. Strengths and Limitations

In our analysis, we found significantly decreased catestatin levels in women with preeclampsia compared to healthy control patients, affirming the presumed association of catestatin with preeclampsia. Furthermore, our data analysis allowed us to identify potential thresholds for ruling in or ruling out preeclampsia. To the best of our knowledge, potential cut-off values have not been described previously. Therefore, we believe this work to be especially valuable. A limitation of this study is the small sample size. However, after performing power analysis and considering this is an exploratory study, we found our sample size to be more than adequate for efficient statistical analysis. Nevertheless, in order to confirm our results or to calculate more precise cut-off values, a larger sample size would be advantageous.

## 5. Conclusions

In conclusion, we demonstrated significantly decreased maternal serum catestatin levels in women with preeclampsia compared to healthy controls. Our results are in line with data showing that circulating levels of catestatin are decreased in patients with chronic hypertension. The role of catestatin as a blood pressure regulator in the development of hypertension has been described in non-pregnant adults. Furthermore, this was the first study to investigate the prognostic effect of catestatin in the prediction of preeclampsia, showing an additive prognostic value and further indicating an association between decreased catestatin values and the presence of preeclampsia. Additionally, we were able to identify potential thresholds for ruling in or ruling out preeclampsia. These results warrant further investigations into the diagnostic value and possible therapeutic effects of catestatin in preeclampsia.

## Figures and Tables

**Figure 1 jcm-12-05931-f001:**
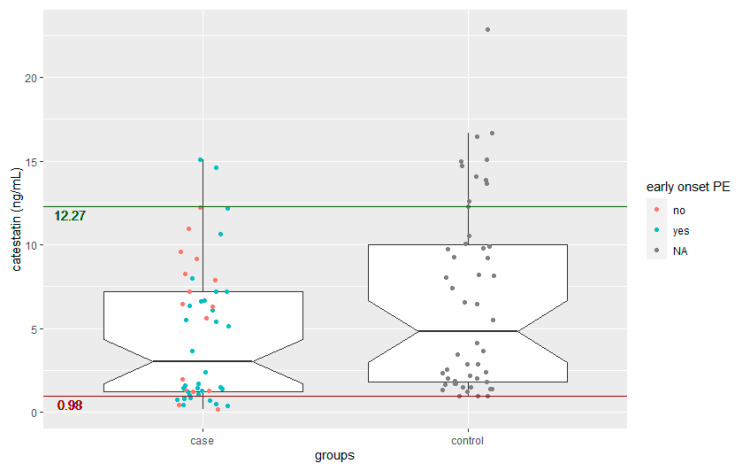
Catestatin levels by group. Notched boxplots for catestatin levels; two boxplots, one for each group, showing median and IQR for catestatin levels. The notches correspond to the 95% confidence intervals for the medians. The scatter plot has been colored (red for late-onset preeclampsia, blue for early-onset preeclampsia and grey for controls). Additionally, two cut-offs are visualized: a catestatin level of 0.98 to rule in preeclampsia (red line) and a catestatin level of 12.27 to rule out preeclampsia (green line). PE: preeclampsia.

**Figure 2 jcm-12-05931-f002:**
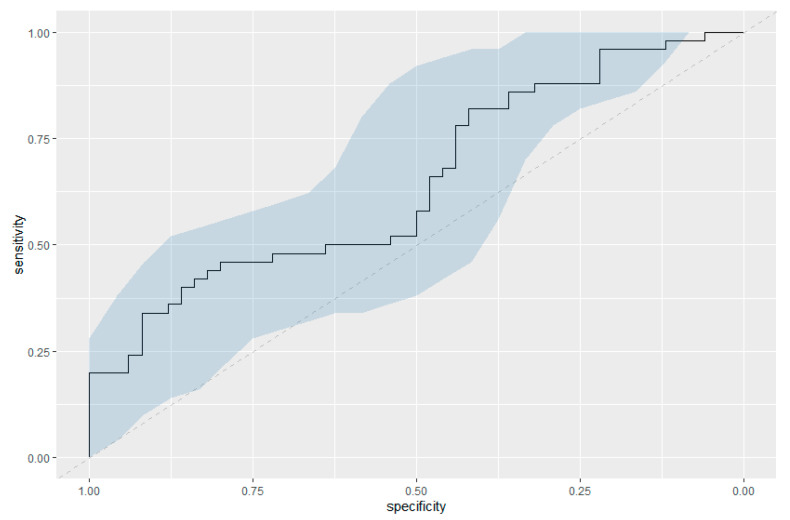
ROC curve for catestatin levels and preeclampsia ROC curve; 95% confidence interval for the curve has been included (blue ribbon).

**Table 1 jcm-12-05931-t001:** Demographics, clinical features and pregnancy outcome of women with and without preeclampsia.

	All (*n* = 100)	Preeclampsia (*n* = 50)	Control (*n* = 50)	*p*-Value
Maternal age in years, median (IQR)	30.5 (26.00–36.00)	33 (27.25–37.00)	29.5 (26.00–34.00)	0.0576
BMI at booking, kg/m^2,^ median (IQR)	27.3 (24.20–31.68)	29.4 (25.88–34.05)	26.05 (23.53–29.08)	0.002
nulliparity, *n* (%)	51 (51%)	32 (64%)	19 (38%)	0.0164
Catestatin in ng/mL, median (IQR)	3.92 (1.43–9.18)	3.03 (1.24–7.21)	4.82 (1.82–10.02)	0.01
MAP at sampling in mmHg, median (IQR)	104.00 (83.50–128.84)	128.00 (114.25–135.5)	83.33 (76.67–87.33)	<0.0001
GA at sampling in weeks, median (IQR)	31.89 (27.36–34.86)	32 (28–34.96)	31.93 (27.04–34.57)	0.607
GA at delivery in weeks, median (IQR)	37.57 (32.89–39.71)	32.79 (28.93–36.11)	39.71 (38.57–40.00)	<0.0001
Delivery before 37 weeks of gestation	42 (42%)	41 (82%)	1 (2%)	<0.0001
SGA, *n* (%)	33 (32%)	31 (62%)	2 (4%)	<0.0001
Birthweight centile, median (IQR)	30.00 (5.00–58.00)	6.00 (2.00–18.75)	52.00 (32.50–76.75)	0.0191
Mode of delivery				
Vaginal delivery, *n* (%)	37 (37%)	3 (9%)	30 (60%)	0.1006
Primary cesarean, *n* (%)	54 (54%)	30 (88%)	14 (28%)	
Secondary cesarean, *n* (%)	9 (9%)	1 (3%)	6 (12%)	
Pregnancy outcome				
Live birth, *n* (%)	96 (96%)	30 (88%)	50 (100%)	0.042
IUD, *n* (%)	1 (1%)	1 (3%)	0 (0%)	
NND < 1 week postpartum, *n* (%)	3 (3%)	3 (9%)	0 (0%)	
NICU admission, *n* (%)	32 (32%)	31 (94%)	0 (0%)	<0.0001
Maternal age in years, median (IQR)	30.5 (26.00–36.00)	33 (27.25–37.00)	29.5 (26.00–34.00)	0.0576

BMI: body mass index; GA: gestational age; IUD: intrauterine death; MAP: mean arterial pressure; NICU: neonatal intensive care unit; NND: neonatal death; SGA: small for gestational age.

## Data Availability

Data available upon reasonable request.

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
