# Peer review of "Catestatin—A Potential New Therapeutic Target for Women with Preeclampsia? An Analysis of Maternal Serum Catestatin Levels in Preeclamptic Pregnancies"

_jcm, 2023, doi:10.3390/jcm12185931_

Round 1

Reviewer 1 Report

In this manuscript, the authors indicated Catestatin - A Potential New Therapeutic Target for Women 2 With Preeclampsia?- Analysis of Maternal Serum Catestatin 3 Levels in Preeclamptic Pregnancies. 

1.     In introduction, After sentences, ‘‘Increased blood pressure remains the main target of therapy trying to reduce maternal and neonatal morbidity and mortality by using antihypertensive agents.’’ It should be given information. For this purpose the authors can look at the following articles for introduction section: Journal of pharmacy and pharmacology 71 (10), 1576-1583

2.     Wouldn't it be better if HDL-LDL ratios were also looked at in the study?

3.     It would be good if it could be discussed with a little more literature in the Discussion section.

4.     Grammaticals errors found in the manuscript. It should be corrected.

In this manuscript, the authors indicated Catestatin - A Potential New Therapeutic Target for Women 2 With Preeclampsia?- Analysis of Maternal Serum Catestatin 3 Levels in Preeclamptic Pregnancies. 

1.     In introduction, After sentences, ‘‘Increased blood pressure remains the main target of therapy trying to reduce maternal and neonatal morbidity and mortality by using antihypertensive agents.’’ It should be given information. For this purpose the authors can look at the following articles for introduction section: Journal of pharmacy and pharmacology 71 (10), 1576-1583

2.     Wouldn't it be better if HDL-LDL ratios were also looked at in the study?

3.     It would be good if it could be discussed with a little more literature in the Discussion section.

4.     Grammaticals errors found in the manuscript. It should be corrected.

Reviewer 2 Report

All corrections should be done before the start of publication process

Conclusion should be more expanded

There is no recommendations

There is no plan for the study area ??

Abstract is very short --why ???

There is a huge a mounts of abbreviations --it will be better if you create a separate table for this

LN/13-14--pathogenesis of preeclampsia---mention in details

Is the decrease in the level of catestatin is an powerful indicator for the presence of preeclampsia or not and why ???

LN/23--add maternal and neonatal morbidity and mortality, Chrmogranin and retrospective analysis to the keywords

LN/34/38/40/43/48---all add references

LN/60---more details are required

LN/62-63---pathogenesis is required

LN/113---There is no reference for the statistical analysis ?

Introduction is extremely very long ---why ??? should be more concise

Some of the  descriptive methodologies are without references --why ???

Are there any relation between catestatin and the PH of the placental fluids and mention its role in induction of preeclampsia ???

LN/14--singleton ---details

LN/32---pathophysiology and pathomechanism---explain ??

LN/41--in 1997---what is the benefit to write the year without the man who discovered ???

LN/43-45--more details are needed for the ordinary readers

LN/47---release of histamine---how and from where

LN/48---animals models--such what ???rats/mice /G pigs/rabbits--etc

LN/48--CgA-knockout mice---explain in detailed manner

LN/150--gestational age---mention it

LN/214-232---this is not a discussion

Discussion is very short--why ??? should be rewrite it again and based upon debating the obtained results with those of the previous investigators results

The authors did not mention anything about the characteristic clinical symptoms of both mother and her baby suffered from preeclampsia

Results should be  more summarize

Discussion is extremely very long ---why??? should be more concise and based upon debating the obtained results with those of the previous investigators results

discussion should be rewritten again

Write as Table(1):----------/Fig.(1):-------etc---apply for all

Some cited references need to be more update

As volume/issue/number/pages---are available---so no for the link(s)----apply for all

Some journal names were written abbreviated , while others were not--why ?? same style should be ---apply for all

Ref(1)--delete vol. ,no. , pp--and write as 387(10022):999-1011/2016---etc---apply for all

Ref(2&3--etc----) we don't add etal with the reference lists unless the total number of authors exceed than 6 ---we add etal with the last ones --apply for all

Round 2

Reviewer 1 Report

The manuscript can be accepted this form.